# Diatoms Biomass as a Joint Source of Biosilica and Carbon for Lithium-Ion Battery Anodes

**DOI:** 10.3390/ma13071673

**Published:** 2020-04-03

**Authors:** Andrzej P. Nowak, Myroslav Sprynskyy, Izabela Wojtczak, Konrad Trzciński, Joanna Wysocka, Mariusz Szkoda, Bogusław Buszewski, Anna Lisowska-Oleksiak

**Affiliations:** 1Chemical Faculty, Gdańsk University of Technology, Narutowicza 11/12, 80-233 Gdańsk, Poland; kontrzci@pg.edu.pl (K.T.); joanna.wer.wysocka@gmail.com (J.W.); mariusz.szkoda@pg.edu.pl (M.S.); 2Faculty of Chemistry, Nicolaus Copernicus University, Gagarina 11, 87-100 Toruń, Poland; mspryn@chem.umk.pl (M.S.); 503151@doktorant.umk.pl (I.W.); bbusz@umk.pl (B.B.)

**Keywords:** biosilica, anode material, lithium-ion batteries

## Abstract

The biomass of one type cultivated diatoms (*Pseudostaurosira trainorii*), being a source of 3D-stuctured biosilica and organic matter—the source of carbon, was thermally processed to become an electroactive material in a potential range adequate to become an anode in lithium ion batteries. Carbonized material was characterized by means of selected solid-state physics techniques (XRD, Raman, TGA). It was shown that the pyrolysis temperature (600 °C, 800 °C, 1000 °C) affected structural and electrochemical properties of the electrode material. Biomass carbonized at 600 °C exhibited the best electrochemical properties reaching a specific discharge capacity of 460 mAh g^−1^ for the 70th cycle. Such a value indicates the possibility of usage of biosilica as an electrode material in energy storage applications.

## 1. Introduction

One of the most critical issues of our time is the escalating climate catastrophe. The cause of climate change has been known since the explanation of the effect of the increase in CO_2_ concentration in the atmosphere on Earth temperature by Arrhenius in the XIXth century [1]. Reduction of anthropogenic CO_2_ emission is demanded and this is indisputable. Arrhenius’ descendant Greta Thunberg, as many other people from all generations, raise the alarm, which should help the local and global authorities in launching measures to slow down the observed changes and to help stop burning fossils. Geothermal energy, wind energy and solar energy, all free, abundant and carbon neutral are most desirable [2]. Photovoltaic technologies are blooming now due to perovskite chemistry, showing potential for carbon dioxide emission slowing down [3]. Wind energy and solar energy require storage. Energy storage systems in the form of batteries or electrochemical capacitors are needed both for dispersed energy and for portable electronic devices [4]. The fabrication of lightweight, high performance batteries and other electrochemical storage devices has the potential to get rid of the transportation paradigm—petrol engine vehicles. Here we focus on lithium ion batteries (LiBs), secondary cells being on the market since 1991. Nobel Prize laureates in chemistry have contributed to the introduction of reversible lithium-ion cells as devices of crucial importance for the functioning of modern societies [5]. Fundamental works of Whittingham and Goodenough were essential in understanding the role of intercalation in battery reactions [6,7,8,9]. Understanding intercalation resulted in the first commercial lithium rechargeable batteries that were built by Exxon. Safety reasons, due to dendrite formation upon charging, forced changes and replacement of the metallic anode to graphite. In this way lithium-ion batteries were introduced by Sony in 1991. We owe the change of the Li metal to graphite to Yoshino, Nobel Prize winner in chemistry for 2019 [5]. Lithium ion batteries, after being three decades on the market, demand better chemistry giving higher performance parameters. The theoretical capacitance of graphite—372 mAh/g is 10 times lower than that of Li metal. The search for safe anodes with higher operating parameters is being undertaken. This is a challenge for material chemistry, nanotechnology and electrochemistry. Among proposed chemistries for next generation anodes there are two groups—one based on alloying (Li_x_Sn, Li_y_Si), the other based on oxides and oxosalts [10,11].

One of the most promising negative electrode materials for next generation anodes is silicon due to its high theoretical specific capacity of 3579 mAh g^−1^ for Li_4.4_Si [10]. The problem with silica is that it undergoes significant volume changes during alloying/dealloying with lithium ions [12]. It leads to material pulverization and loss of electric contact between the active material and the current collector. However, the advantage of silicon utilization is that it is one of the most abundant elements on Earth existing in the form of SiO_2_ of different origins. Although the theoretical capacity of silica is lower than for silicon, the material itself is very attractive for energy storage and energy conversion application [13,14,15].

Here presence of silicate/silica can buffer volumetric changes of the material under polarization. Theoretical charge capacitance is very attractive as being higher than that for graphite. However, silica, as a non-conductive material, should be ground to nanometric sizes so that Faradaic reactions can occur. Materials chemistry provides nano-silica in the form of an aerogel active reversibly as the anode if a carbon nano-coating is created on top of SiO_2_ properly [16]. Another example of obtaining silica nanomaterial appropriate for anode application is based on mechanical grinding of beach sand [17]. Ground sand, as in the case of the SiO_2_ aerogel, is subjected to the procedure of forming a composite with an electron conductor—sp^2^ carbon, often as acetylene black. Nanosilica uniformly dispersed within the electron conductor should act efficiently, combating volumetric changes of the alloying reaction. The most crucial for the charge transfer reaction is the quality of the interface between carbon and silica particles. Here, the cation Li^+^ and the electron are taking part in the faradaic process.

In this study we focus on a biological source of both silica and carbon in one living species. We chose diatoms giving jointly biosilica and carbon precursors, (like “two in one” items). The ability of diatoms to make silica shells has recently attracted the attention of materials chemists and molecular biologists, leading to a rapid increase in our understanding of the biochemistry of silicification [18]. Strikingly beautiful is the elaborated structure and regular pattern of ribs and pores promoted by protein-bound polyamines in the silicification process [18,19]. The bare silica skeleton of diatoms frustules, chemically stripped of proteins, peptides, polysaccharides, is described in ref. [20]. The effects of mechanical stress, resulting from alloying, is the major failure causing loss of electrical contact and can be tackled mechanically [21]. In addition, confinement/solvation effects of lithium ions plays a decisive role in efficiency of lithium-ion batteries [22,23].

In the case of diatomic silica this malfunctioning can be minimized through the intricate construction of diatom frustules. Evenly repeating elliptical gaps/holes create a system limited by a coherent structure of biosilica spans, Figure 1.

The structure with convex ribs helps to counteract mechanical stress (as the designers of the Gothic vaults used convex spans effectively). Diatoms architecture causes that under grinding tests frustules are elastic and resilient [24,25]. This feature is expected to help maintain the mechanical stability of the electrode material under lithiation of the silica part of diatoms. Mechanical properties of the diatoms structure attracted attention of Greer et al. [26]. Diatoms BioSilica (DB) skeleton is covered with matter rich in carbon compounds [27,28]. Here our aim is to take advantage of these species and use them as a precursor of sp^2^ carbon—an electron conductor. In our previous works [20,29,30,31] we have shown that biosilica is electroactive and stable during lithiation. All tests clearly show that diatoms frustules possess potential for practical use in LiBs [32]. Recently work of Norberg et al. confirmed applicability of silica frustules from sea-hauled coscinodiscus as anode material for lithium batteries with good gravimetric capacity equal to 723 mAh g^−1^ for composite and 624 mAh g^−1^ after additive subtraction [33].

However, there are still questions to be answered, the first is determination of cultivation conditions on electrochemical performance of the obtained biomass, focusing on both the skeleton DB and the carbon part derived from organic components. Using language to popularize science, one can define diatom as a 2-in-1 object for present application in LIBs as an anode. The goal of this work is to prove that by using what Nature has created, we are cutting costs in the technological process to obtain a SiO_2_@C nanostructure. Firstly, due to diatom biosilica being a kind of nanostructure, further downsizing of the electrode material is not required. Secondly, closely attached to silica organic parts can be easily thermally turned to carbon, so additional carbon precursors are unnecessary.

Here we show tests performed on one type of diatom class *Pseudostaurosira trainorii*. After thermal treatment the biomass was subjected to structural and electrochemical studies to elucidate electroactivity of biosilica imbedded in a carbonaceous matrix. Cultured diatoms were pyrolysed at various temperatures and used as an anode material for LiBs.

## 2. Materials and Methods

Diatom biomass was obtained by cultivation of the selected diatom species (*Pseudostaurosira trainorii*) under laboratory conditions using Erlenmeyer flasks with with Guillard’s nutrient solution F/2 (Merck, Germany) containing silicon at a concentration of 7 mg dm^−1^ as potassium metasilicate. The cultivation was carried out at light/dark regime of 12:12 h, aeration and a constant temperature 22 °C. After the growing time the diatom biomass was separated by decantation, washed with distilled water using centrifugation and dried at 70 °C.

The diatom biomass (DB) was filled in a ceramic crucible, put into a quartz tube, evacuated, subsequently filled with argon and finally heated under a steady flow of argon (Ar 5.0) (25 mL min^−1^) in a programmable horizontal tube-furnace (Czylok, Poland) with a heating rate of 100 °C h^−1^ to the final temperature (600 °C (named DB@C at 600 °C), 800 °C (named DB@C at 800 °C) and 1100 °C (named DB@C at 1100 °C)) and held at the final temperature for 2 h.

The pyrolysed samples were mixed with the binder (polyvinylidene fluoride PVdF, Solef 6020, Rheinberg Germany) and the conducting additive (Carbon Black Super P^®^, Timcal Ltd., Bodio, Switzerland) in NMP (Avantor Performance Materials Poland S.A., Gliwice, Poland). The weight ratio of active material to the binder and to the conducting additive was adjusted as 90:5:5 by weight. The slurry was homogenized in the ball mill (Mixer Mill MM200, Retsch, Haan, Germany) and was cast on the rough side of a 10 μm thin copper foil (kindly donated by Schlenk Metallfolien GmbH & Co. KG, Georgensgmünd, Germany) and spread out with a metal hand blade. After the tape casting procedure, the electrode strip was dried for 8 h at 100 °C in an oven (Glass Oven B-585 Büchi, Essen, Germany). Then the disks of 10 mm diameter (up to 2 mg of material) were cut out of the tape and pressed for 30 s under a pressure of 950 MPa (9.7 × 10^9^ mg cm^−2^). After cutting the disks were dried under dynamic vacuum in an oven (Glass Oven B-585 Büchi, Germany) for 24 h at 80 °C. Such disks served as a working electrode in a two-electrode Swagelok type cell. The lithium foil (99.9% purity, 0.75mm thickness, AlfaAesar) was used as the counter and reference electrode for the two-electrode half-cell configuration. A glass-microfiber filter (Schleicher & Schüll, Germany)) and 1 M of LiPF_6_ in EC:DMC 1:1 (LP30 Merck, Germany) were used as the separator and the electrolyte, respectively.

The scanning electron microscopy with a focused Ion Beam (SEM/FIB—Quanta 3D FEG) and scanning electron microscopy (SEM, LEO 1430 VP, Leo Electron Microscopy Ltd, Cambridge, United Kingdom) coupled with an energy dispersive X-ray (EDX) detector (XFlash 4010, Bruker AXS) were used to study the morphology features and elemental composition of the pyrolysed diatom biomass.

X-ray powder diffraction (XRD, Malvern Panalytical Ltd, Malvern, United Kingdom) analyses were performed using a Philips X’Pert Pro diffractometer with Cu-Kα radiation (λ = 0.1541 nm, 40 kV, 30 mA). XRD pattern data were collected over an angular range of 5°–120° 2θ with step sizes of 0.01.

The structural bonding and functional groups of the diatom biomass were recorded using an Fourier Transform Infrared (FTIR) spectrophotometer (FTIR ATR, Vertex 70, Bruker Optik, Bremen, Germany) equipped with a DLaTGS detector and a Dispersive Raman spectrometer Senterra (Bruker Optik, Rosenheim, Germany). The FTIR spectra were recorded by averaging 64 scans in the wave number range 400 cm^−1^ to 4000 cm^−1^ with a resolution of 4 cm^−1^.

The thermal behaviour of the diatom biomass was investigated in the temperature range 20–1100 °C using a TGA–DTA Thermal Analysis Instruments SDT 2960 derivatograph (TA Instruments, Warszawa, Poland). The samples were heated at a heating rate of 5 °C min^−1^ in an air atmosphere.

## 3. Results and Discussion

The morphology features of the pyrolyzed diatom biomass are demonstrated by the scanning electron microphotographs in Figure 2a–c. Figure 1a shows the diatom biomass after heat treatment at 600 °C. The diatom cells after heat treatment are well preserved and remain in the form of colonial ribbons connected to each other by siliceous spines. The diatom cells are exhibited by oval shaped valves with an average diameter of nearly 4–5 μm. It can be observed that the surface of the cells was perforated by periodic bilaterally symmetric rows of oval pores of nanometric sizes (150–200 nm) as it was evidenced by Sprynskyy et. al. in [34]. The residuals of degraded organic matter of diatoms cells after thermal treatment were not deposited separately, but similarly as thin layers (coats) cover the silica exoskeletons of diatom cells. One can also observe mineral inclusions of calcium carbonate (identified by XRD analysis) in the form of the crystallite aggregates. The sizes of the crystallite aggregates were in the range 5–20 mm, and calcium carbonate crystallites were of submicrometer sizes. The microphotographs of diatom biomass after pyrolysis at 800 °C are presented in Figure 2b. The morphology of diatom cells after heat treatment at a temperature of 800 °C did not change and looked like it did after treatment at the temperature of 600 °C. The mineral inclusions were also observed but crystallites with cleaner crystalline forms in mineral aggregates of calcium oxides were not identified. Significant changes in the morphology and structure of the diatom cells can be noted after treatment at a temperature of 1100 °C (Figure 2c). The diatoms cells looked transparent and partially melted or dissolved.

The elementary composition of the diatom biomass after pyrolysis at different temperatures obtained using SEM-EDX is presented in Table 1. These data demonstrated that oxygen, carbon, silicon and calcium were the main elements of the diatom biomass. One may see that after annealing the diatom at 1100 °C, the elemental composition changed significantly. We observed a loss in carbon, oxygen and silicon content. It is very likely that gaseous products of SiO and CO were formed as it is claimed by Biernacki and Wotzak [35]. Admixtures of iron, phosphorus and potassium were also significant.

X-ray powder diffraction patterns of the diatom biomass dried at 50 °C and diatom biomass pyrolysed at 600 °C, 800 °C and 1100 °C are shown in Figure 3. The XRD patterns obtained for dried diatom biomass and pyrolysed diatom biomass at 600 °C exhibited the same distinct crystalline peaks located at around 2 theta = 29.35°, 36.04°, 39.49°, 43.25°, 47.30°, and 48.65°. The detected crystalline peaks were identified as characteristic diffraction peaks of calcium carbonate (Ref. code: 00-001-0837, Calcite, CaCO_3_). This indicates that the calcium carbonate mineralization in the diatom biomass already took place at the stage of diatom biomass drying and the stability of calcium carbonate during pyrolysis processes at 600 °C.

The XRD pattern of the diatom biomass pyrolysed at 800 °C shows the pronounced peaks (2 theta: 32.19°, 37.36°, 53.85°, 64.13°) that are indicating formation of the calcium oxide phase (Ref. code: 00-004-0777). The thermal treating of diatom biomass at 800 °C caused a totally thermal decomposition of calcite CaCO_3_ to calcium oxide CaO.

The diffraction peaks at 32.17°, 32.65°, 32.65°, 39.67° and 41.38° in the XRD spectrum of pyrolysed diatom biomass at 1100 °C are consistent with the crystalline peaks of calcium silicate oxide according to the standards of JCPDS (Ref. code: 00-003-0753, Ca_2_SiO_4_). The formation of calcium silicate can be explained through a solid–solid reaction between the CaO phase and the amorphous SiO_2_·*n*H_2_O of diatom frustules during pyrolysis processes [36]. It is also observed that crystalline peaks corresponded to cristobalite (Ref. code: 00-004-0379, Cristobalite, SiO_2_). The characteristic main peaks of this mineral appeared at 21.98° and 36.05°. The transformation of the amorphous diatom biosilica onto crystalline silica as cristobalite can be testified. The broad peaks at 15–30° 2 theta that appeared on XRD spectra of dried diatom biomass and diatom biomass pyrolyzed at 600 °C and 800 °C may be assigned to amorphous silica such as opal—A [37].

The FTIR spectra of the tested diatom biomass are shown in Figure 4. The FTIR spectrum obtained for not thermally treated biomass displays the bands that are reflected from organic and mineral components. The bands at 2916, 2852, 1737, 1651, 1537, 1467, 1393 cm^−1^ reveal the functional groups of organic compounds and bands at 1439, 1071, 993, 869, 798, 711, 575, 487, 457 cm^−1^ are related to functional groups of the mineral part of the diatom biomass matrix. The band at 1537 cm^−1^ is only seen for pure biosilica and could be assigned to the bending vibration of the N-H groups [38].

The absorption bands at 2916 and 2852 cm^−1^ were assigned to C–H asymmetric and symmetric stretching of aliphatic groups, while the bands at 1467 cm^−1^ and 1393 cm^−1^ were associated with C–H bending vibrations of aliphatic groups of the amines. The band near 1737 cm^−1^ could be attributed to the C=O stretching vibration of the ester carbonyl group from amino acids. The weak band near 1651 cm^−1^ was assigned due to C=O bonds of primary amide groups [39]. The FTIR spectrum of the untreated diatom biomass showed characteristic bands for amorphous biosilica (SiO_2_) and calcium carbonate (CaCO_3_). The bands at 1439 and 869 cm^−1^ corresponded to C–O stretching and bending vibrations, and the band at 711 cm^−1^ was related to Ca–O bonds in the structure of CaCO_3_ [40]. The detected bands at 1071, 798 and 457 cm^−1^ may be related to stretching and bending vibrations of Si–O bonds in the structure of diatom biosilica [41,42].

The FTIR spectrum obtained for biomass pyrolysed at 600 °C does not show characteristic bands coming for organic compounds and indicates the total destruction of the organic component. Furthermore, mineral phases (biosilica and calcium carbonate) were not transformed to other forms, while, the changes in the FTIR spectrum that are related to changes in the mineral phase can be observed for the diatom biomass pyrolysed at 800 °C. In this case, the bands belonging to calcium carbonate (869 cm^−1^ and 711 cm^−1^) disappear and a new narrow band at 3642 cm^−1^ appears. This indicates transformation of calcium carbonate to calcium oxide (CaO) [43]. After thermal treatment of diatom biomass at 1100 °C new bands appear in the FTIR spectrum in the 1000–450 °C range at 993, 897, 609, 564 and 497 cm^−1^ that are associated with dicalcium silicate mineralization (Ca_2_SiO_4_), wherein the bands at 993 and 897 cm^−^1 are attributed to Si-O-Ca stretching vibrations of dicalcium silicate [44]. The mineral transformation course in the order CaCO_3_-CaO-Ca_2_SiO_4_ of thermal treatment of diatom biomass in the 1000–450 °C temperature range was confirmed also by the XRD analysis.

The data on thermal stability and phase transformation of the diatom biomass that was obtained by thermogravimetric analysis (TG—thermogravimetric, DTG—derivative thermogravimetric, and DTA—differential thermal analysis) are shown in Figure 5. Five different phases of mass loss can be identified in the thermogravimetric curve. The first phase with a weight loss of about 2% manifested in the temperature interval 25–105 °C. This weight loss was related to dehydration processes (release of physically bonded water) of the biomass accompanied by an endothermic effect on the DTA curve and a weak DTG peak. The second phase appeared in the temperature interval 150–420 °C with weight reduction of about 25%, an exothermic effect and a DTG peak centred at a temperature of 260 °C. This phase was related to the decomposition of organic components. The third phase occurred in the temperature interval 605–705 °C with an endothermic effect and weight loss of about 22%. This phase was characterized by a strong asymmetric DTG peak and may be assigned to destruction of calcium carbonate accompanied by carbon dioxide emission [36]. The fourth phase was separated in the temperature interval 705–850 °C with a gradual weight loss of 5%. This phase corresponded to a dehydroxylation process and formation of calcium oxide. The fifth phase manifested at a temperature above 950 °C by a clearly exothermic effect of the DTA curve without weight loss and could be referred to the formation by calcium oxide reacting with silicon oxide of dicalcium silicate [36,45].

The Raman spectra of diatomic biosilica embedded in a carbonaceous matrix (DB@C) are shown in Figure 6. One may see that the most intensive peaks originated from the carbon phase and are recorded with two main maxima in the range from 1000 cm^−1^ to 1800 cm^−1^. This region is characteristic for carbonaceous materials of both: ordered [46,47] and disordered structures [48,49]. Such materials exhibit two broad and overlapping peaks with intensities located around ~ 1350 cm^−1^ and 1580 cm^−1^. The first is known as D mode (“disordered”) and is attributed to the breathing motion of the sp^2^-ring. The G mode (“graphite”) resulted from an in-plane bond stretching of sp^2^ carbon atoms. Both modes evidenced the presence of an aromatic ring in the material structure [50]. The presence of disorder in the carbon phase expects one to take into account several first-order Raman bonds for a complete analysis and interpretation [51]. In Figure 6b there is an exemplary deconvolution curve with five bands for DB@C at 600 °C for the electrode material. The D1, D and D3 bands are attributed to the disordered graphitic lattice while the D2 band origins from the presence of an amorphous phase typical for soot materials.

The G band confirms the presence of an ideal graphitic lattice (see Table 2). A particle size of crystalline carbon domains (*L*_a_) was calculated with the formula given by Ferrari and Robertson for amorphous carbonaceous materials [52]:(1)$$I\left(D\right)/I\left(G\right)=C\prime \left(\lambda \right)\cdot L_{a}^{2}$$
where *C*′ (for 514 nm) = 0.0055 Å^−2^, and *I(D)/I(G)* is the intensity ratio between D and G bands.

The cluster size slightly decreased from 2.2 nm to 1.7 nm with the carbonization temperature. This constant value evidenced that for the applied pyrolysis temperature the carbon phase underwent structural changes. The G bands’ full width at half maximum (FWHM) changed with the temperature increase showing values 56.8 cm^−1^ for 600 °C, 52.3 cm^−1^ for 800 °C and 39.2 cm^−1^ for 1100 °C. This narrowing of the G peak evidenced an increase in the crystalline order with increase of the pyrolysis temperature.

The summarized data of the bands position and the *L*_a_ parameter value are gathered in Table 3.

The presence of biosilica was confirmed by the maxima in the range below 1100 cm^−1^ originating from Si-O and Si-O-Si vibrations. One may see that the positions of the peaks are not detected at the same locations. The common feature of Raman spectra of all electrode materials wasa small hump at 435 cm^−1^ which may arise from symmetrical Si-O-Si stretching modes involving motion of oxygen atoms [53]. The material pyrolyzed at 600 °C exhibited two sharp maxima at 1085 cm^−1^ and 284 cm^−1^. The former might be attributed to Si-O-Si asymmetric stretching [54] and the latter is very likely assigned to a sum of high energy (E) longitudinal optical (LO) and transverse optical (TO) phonon coupled modes (E_LO+TO_) [55]. We were not able to detect any visible signals, except the one at ~ 440 cm^−1^, confirming the presence of biosilica in DB@C at 800 °C. For material carbonized at 1100 °C three sharp maxima were seen at 973 cm^−1^, 571 cm^−1^ and 368 cm^−1^. The first is assigned to the surface Si-OH stretching mode [56]. The second might be attributed to the D_2_ line assigned to the symmetric oxygen breathing vibration of the three-membered siloxane ring, consisting of SiO_4_ tetrahedra in silica [57]. The third band is assigned to the structural lattice mode in silica [58].

The differences in bands positions suggest that biosilica underwent structural changes during thermal treatment [59]. The presence of the source of carbon in pristine biosilica might also be affected by the position of Si-O maxima. It is known that SiO_2_ may react with C to form SiC [60]. Although we did not reach temperatures allowing for carbothermic reduction of silica it is very likely that the distance between Si and O was changed.

Cyclic voltammetry curves of the electrode material consisting of diatomic biosilica pyrolysed at different temperatures are shown in Figure 7. One may see that the shape of the curves is similar for all of them. In the first scan there is a broad hump at 1.1 V and a cathodic maximum at ~0.7 V. The former hump is very likely attributed to the reaction with silica followed by non-stoichometric lithium silicate formation. The maximum at 0.7 V vanishes in the second and third cycle. It evidences formation of a solid electrolyte interphase (SEI)—a surface layer which is formed by the reaction of the metal with the electrolyte, and has the properties of a solid electrolyte [61,62]. Lithium insertion into the electrode material begins at a potential below 0.30 V. This process includes both lithium intercalation to the graphitic lattice [63] and a lithium reaction with SiO_2_ [64]:5SiO_2_ + 4Li^+^ + 4e^−^ ↔ 2Li_2_Si_2_O_5_ + Si(2)
SiO_2_ + 4Li^+^ + 4e^−^ → 2Li_2_O + Si(3)
2SiO_2_ + 4Li^+^ + 4e^−^ → Li_4_SiO_4_ + Si(4)
Si + xLi^+^ + xe^−^ ↔ Li_x_Si(5)

Among above mentioned reactions only reactions (2) and (5) are reversible. This causes the theoretical reversible capacity of SiO_2_ to be in the range from 749 mAh g^−1^ to 1961 mAh g^−1^ depending on the mechanism.

One may see that there is a redox couple activity at the potential ~1.0 V. Such activity was also observed by Zhang [58] and Chen [60] for SiO_2_ system. However, the origin of this activity was not clearly explained by those authors. We suppose that it is attributed to delithiation of lithium silicate as was described in [16].

Figure 8 shows the relation between the specific capacity and cycle number of DB@C thermally treated at different temperatures and at different current densities.

One could see a high irreversible capacity after the first charging process. This capacity loss was due to SEI formation, as well as due to the irreversible reaction between lithium-ions and silica followed by lithium oxide and lithium silicates formation. Moreover, during cycles from 1 to 5 at current density j = 40 mA g^−1^ there was a noticeable difference in specific capacities between charging and discharging processes. The specific capacity of charging was higher than for discharging. We claimed that this phenomenon was attributed to formation of a stable form of silica for lithium ion insertion. Much of it was correlated with irreversible reactions (2) and (3). Pyrolysis of diatomic bisilica containing an organic source of carbon was expected to prevent such behavior by incorporation of DB into the carbonaceous matrix. However, one may see that we did not succeed for the first cycles. Nevertheless, all materials exhibited stable electrochemical behaviour for higher current densities. There is a relationship between the specific capacity and the pyrolysis temperature. One may see that the material carbonized at 600 °C showed a higher capacity value in comparison with the material thermally treated at 800 °C and 1100 °C. After the 65th cycle at j = 500 mAh g^−1^ the specific capacity was equal to 240 mAh g^−1^, 160 mAh g^−1^ and 160 mAh g^−1^ for pyrolysis at 600 °C, 800 °C and 1100 °C, respectively. It is noteworthy that after decreasing the current rate initial value (40 mA/g) the capacity increased. The discharge capacity after 70 cycles was 460 mAh g^−1^ for 600 °C, mAh g^−1^ for 800 °C and 320 mAh g^−1^ for 1100 °C. The specific capacity value for DB@C at 600 °C was higher than for the graphite electrode (372 mAh g^−1^). The decrease in capacity for material carbonized at higher temperatures might by due to structural changes of the electrode material confirmed by XRD as formation of CaO and Ca_2_SiO_4_, see Figure 2. The presence of both CaO in DB@C at 800 °C and Ca_2_SiO_4_ in DB@C at 1100 °C affected the total mass of the electrode material and have an impact on the calculated specific capacities. Thus, the specific capacity of DB@C at 800 °C and DB@C at 1100 °C was lower in comparison with the DB@C at 600 °C electrode material.

The summarized data on electrochemical results is gathered in Table 4.

## 4. Conclusions

Laboratory cultivated diatom algae of one selected type *(**Pseudostaurosira trainoriic)* were chosen as a source of electroactive material precursor. The aim of this study was to use the whole biomass as a joint source of carbonaceous material- electron conductor and silica—the main electroactive chemistry used as an anode material for lithium ion batteries.

Here it is demonstrated that the pyrolysis temperature of biomass leads to formation of electroactive matter obtained at 600, 800 and 1100 °C. It is proven that the technological stage of diatoms chemical treatment, performed to remove the organic part, can be omitted. Moreover, addition of carbon black to the electrode film can be reduced to a very low level (~5%). It was demonstrated, using electrochemical and solid-state physics techniques, that the pyrolysis temperature strongly influences the electrode activity of novel materials. The best results in respect to charge capacity were reached for DB600—obtained at the lower temperature with a specific capacity of 460 mAh g^−1^ after the 70th cycle at a current density of 40 mA g^−1^. Applying a higher current density (0.5 A g^−1^) showed the specific capacity equal to 240 mAh g^−1^. It confirmed that the redox reaction is the slowest process in the system suggesting utilization of diatom biosilica in large-scale energy storage when fast charging/discharging is not required. The main disadvantage in electrode performance results from the type of nutrition and agglomeration of calcium carbonates in the biomass. Almost half of the weight of DB600 comes from calcium compounds mainly in the form of CaCO_3_ at 600 °C, turned to CaO at 800 °C and further to dicalcium silicate at 1100 °C with loss of the carbon phase in the obtained ceramic matter. Calcium carbonate, carbon oxide and Ca_2_SiO_4_ are not expected to be electroactive. Fortunately in the case of CaCO_3_ the negative effect is seen rather as an additional weight added, electrochemical polarization curves exhibit a classical view of SiO_2_@C activity. However, formation of calcium silicate causes diminution of the active silica amount. To overcome this disadvantage one should reduce the amount of calcium compounds in the cultivated biomass. Further improvement in application of diatoms as a source of 2-in-1, namely carbon and silica, is possible and planned.

Concluding, the obtained results are very promising and future work should involve changes in the nutrition procedure to reach a low calcium level in cultivated species.

## Figures and Tables

**Figure 1 materials-13-01673-f001:**
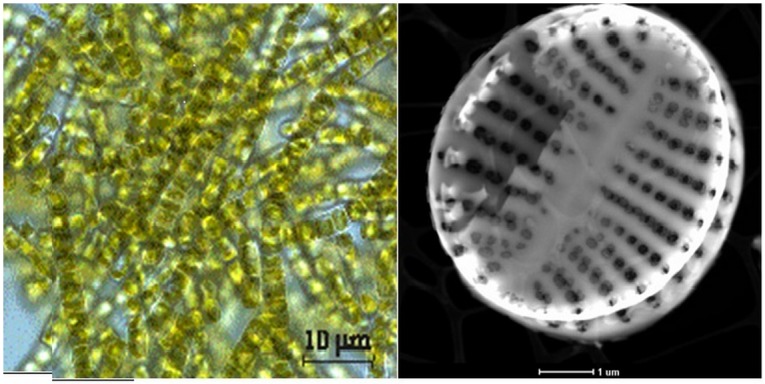
Cultivated diatoms, light microscopy image (left), bare diatom frustules stripped carbonaceous species chemically (right), TEM image.

**Figure 2 materials-13-01673-f002:**
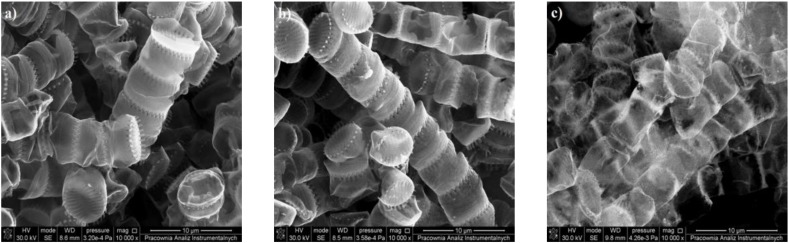
Scanning electron microscopy images of the diatom biomass carbonized at different temperatures (**a**) 600 °C (**b**) 800 °C and (**c**) 1100 °C.

**Figure 3 materials-13-01673-f003:**
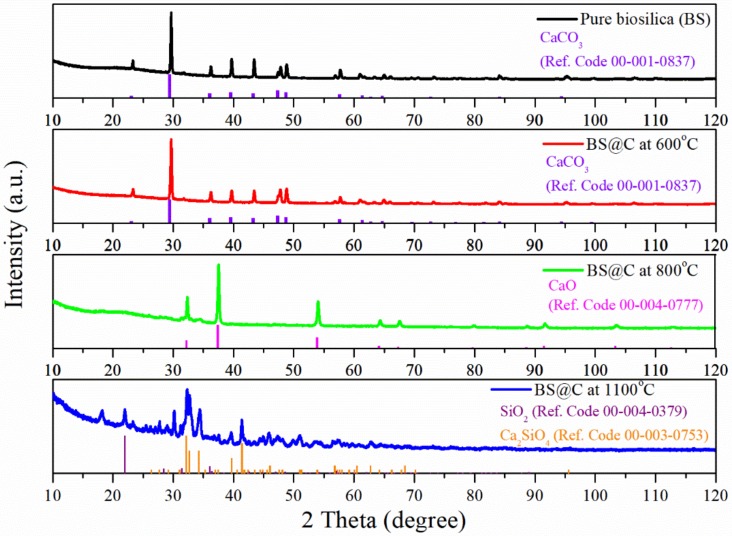
Powder X-ray diffraction patterns of the dried diatom biomass and diatom biomass pyrolysed at 600 °C, 800 °C and 1100 °C.

**Figure 4 materials-13-01673-f004:**
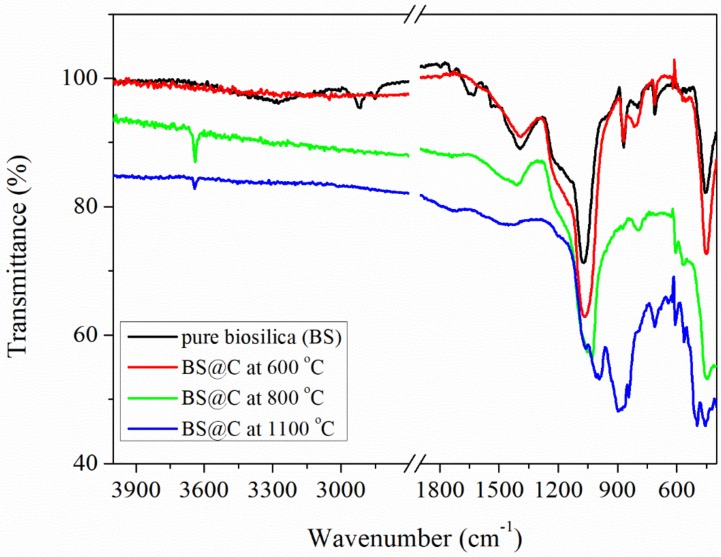
Comparative FTIR-ATR spectra of the dried diatom biomass (**⸺**) and diatom biomass pyrolyzed at 600 °C (**⸺**), 800 °C (**⸺**) and 1100 °C (**⸺**).

**Figure 5 materials-13-01673-f005:**
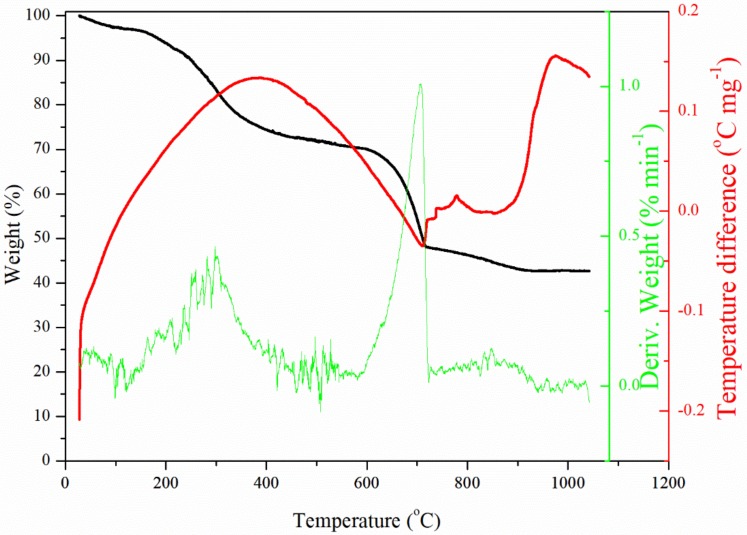
Phase transformations and thermal stability of the dried diatom biomass.

**Figure 6 materials-13-01673-f006:**
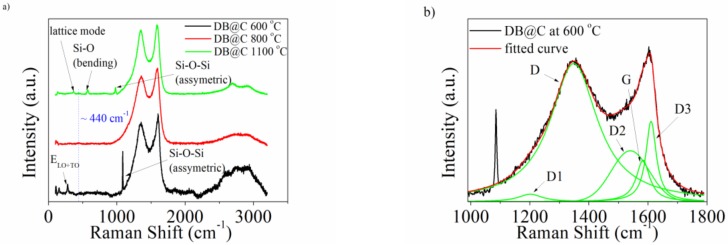
(**a**) Raman spectra for DB@C electrode materials and (**b**) the curve fit for first-order Raman spectra of DB@C at 600 °C.

**Figure 7 materials-13-01673-f007:**
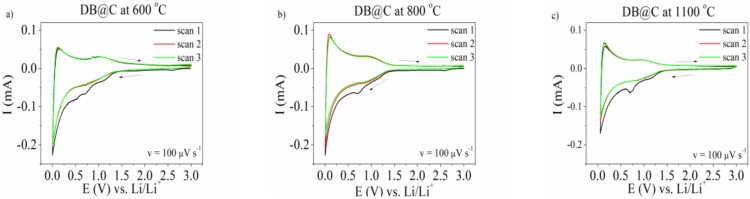
The cv curve of DB@C electrode material pyrolysed at different temperatures (**a**) 600 °C, (**b**) 800 °C, and (**c**) 1100 °C in 1M LiPF6 in EC/DMC. Potential range 0.002–3 V. Sweep rate v = 100 μV s^−1^.

**Figure 8 materials-13-01673-f008:**
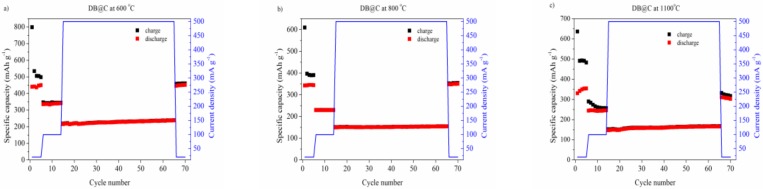
The capacity vs. cycle number of diatom biosilica electrodes at different current densities for (**a**) DB@C at 600 °C, (**b**) DB@C at 800 °C and (**c**) DB@C at 1100 °C.

**Table 1 materials-13-01673-t001:** Elemental composition (wt.%) of the diatom biomass pyrolysed at 600 °C, 800 °C and 1100 °C.

Treatment Diatom Biomass	C	O	Si	Ca	P	Fe	S	Mg	K	Cl
600 °C	24.79	37.14	15.12	18.37	1.85	1.26	0.07	0.25	1.08	0.07
800 °C	22.12	36.53	18.30	18.69	1.19	1.46	0.18	0.34	1.13	0.06
1100 °C	15.14	29.01	12.37	39.77	1.59	1.01	0.15	0.48	0.38	0.01

**Table 2 materials-13-01673-t002:** First-order Raman bands and vibrational modes.

Band	Raman Shift (cm^−1^)	Vibration Mode
D1	~1200	In-plane, due to disorder or/and impurities (A_1g_ symmetry)
D	~1350	Disordered graphitic lattice (A_1g_ symmetry)
D2	~1500	Amorphous/diamond like carbon (sp^3^ hybridized carbon)
G	~1500	In-plane, C-C stretch. (E_2g_ symmetry)
D3	~1610	Disordered graphitic lattice (E_2g_ symmetry)

**Table 3 materials-13-01673-t003:** The band position and *L*_a_ parameters for DB@C pyrolysed at different temperatures.

Electrode Material	Band (cm^−1^)					*L*_a_ (nm)
	D1	D	D2	G	D3	
DB@C at 600 °C	1201	1349	1540	1583	1610	2.2
DB@C at 800 °C	1195	1347	1529	1578	1609	2.0
DB@C at 1100 °C	1183	1347	1535	1581	1611	1.7

**Table 4 materials-13-01673-t004:** Summarized data of the final discharge capacity at different current densities.

Material	Final Discharge Capacity at Current Density			
	40 mA g^−1^	100 mA g^−1^	500 mA g^−1^	40 mA g^−1^
DB@C at 600 °C	450	340	240	460
DB@C at 800 °C	350	240	160	350
DB@C at 1100 °C	340	230	150	320

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
