# Peer review of "Diatoms Biomass as a Joint Source of Biosilica and Carbon for Lithium-Ion Battery Anodes"

_materials, 2020, doi:10.3390/ma13071673_

Round 1

Reviewer 1 Report

The manuscript reports on the use of biomaterials to form battery electrodes. It is an interesting topic. The presented manuscript add some further knowledge to the field, and therefore I suggest its acceptance for publication after some minor revision. My comments and questions: 

Regarding the SEM images in Fig 2. the authors write about 150-200 nm large pores. It is difficult to see whether the size of these changes with the annealing temperature. I suggest taking images at larger magnifications.

After annealing the diatom at 1100 °C, the elemental composition changes significantly. We observe a loss in carbon and silicon content – what happens to these, or what is the reason of this decrease?

Regarding Figure 3., the cyan colored fonts are very difficult to read (at least for me). I suggest changing it to some other color.

Figure 4.: The title of the y-axis is misspelled (a “t” is missing). The authors show the full spectra, even though any meaningful data is only seen below 2000 1/cm, and above 3000 1 / cm. I suggest including a break in the x-axis.

The charge capacity of the materials decrease with the annealing temperature. In the conclusions however the authors state “Almost half of the weight of DB600 comes from calcium compounds mainly in the form of CaCO3at 600 oC, turned to CaO at 800 oC and further to dicalcium silicate at 1100 oC with loss of the carbon phase in the obtained ceramic matter. Calcium carbonate, carbon oxide and Ca2SiO4are not expected to be electroactive.” From this, I would expect the materials treated at 1100 °C to be the best, but it is not the case. The authors should discuss this in detail!

Author Response

Dear Reviewer,

Please see our response in the attached file.

In behalf of authors

Andrzej Nowak

Reviewer 2 Report

The article is very interesting. The theme of the work is current and has great potential for application.

Author Response

(The authors gave the same response as above.)

Reviewer 3 Report

In this submission to Materials, the authors present an experimental study on 3D-stuctured biosilica and organic matter for use as an electroactive material in a potential range adequate to become an anode in lithium ion batteries. The authors characterize these materials with XRD, Raman, TGA and show that the pyrolysis temperature affected structural and electrochemical properties of the electrode material.

I consider this manuscript to be of interest to the Materials and battery community, and I am supportive of publication with a minor note. There has been recent work by the Guo group who has also used XRD and Raman techniques to characterize new materials for lithium batteries and showed that confinement/solvation effects can also play a decisive role in their efficiency:

ACS Nano, 12, 9775-9784 (2018)
Journal of Physical Chemistry Letters, 9, 1739-1745 (2018)

In particular, these prior works have experimentally shown that confinement effects (similar to that of the diatom) play a critical role in Li-ion mobility and efficiency. With this very minor note, I am supportive of publication.

Author Response

(The authors gave the same response as above.)
